# Motivation and Hesitancies in Obtaining the COVID-19 Vaccine—A Cross-Sectional Study in Norway, USA, UK, and Australia

**DOI:** 10.3390/vaccines11061086

**Published:** 2023-06-10

**Authors:** Janni Leung, Daicia Price, Caitlin McClure-Thomas, Tore Bonsaksen, Mary Ruffolo, Isaac Kabelenga, Gary Lamph, Amy Østertun Geirdal

**Affiliations:** 1Faculty of Health and Behavioural Sciences, The University of Queensland, Brisbane, QLD 4072, Australia; j.leung1@uq.edu.au (J.L.); caitlin.mcclurethomas@uq.net.au (C.M.-T.); 2School of Social Work, University of Michigan, Ann Arbor, MI 48109, USA; mruffolo@umich.edu; 3Department of Health and Nursing Science, Faculty of Social and Health Sciences, Inland Norway University of Applied Sciences, 2418 Elverum, Norway; tore.bonsaksen@inn.no; 4Department of Health, Faculty of Health Studies, VID Specialized University, 4024 Stavanger, Norway; 5Department of Social Work and Sociology, School of Humanities and Social Sciences, University of Zambia, Lusaka P.O. Box 50110, Zambia; isaac.kabelenga@unza.zm; 6School of Mental Health and Learning Disability Nursing, Edgehill University, Lancashire L39 4QP, UK; gary.lamph@edgehill.ac.uk; 7Department of Social Work, Child Welfare and Social Policy, Faculty of Social Sciences, Oslo Metropolitan University, 0167 Oslo, Norway; amyoge@oslomet.no

**Keywords:** vaccination, acceptance, refusal, immunization, uptake, influences

## Abstract

Background: Vaccinations protect the public against serious diseases or death; however, some individuals are hesitant in obtaining them. We aim to contribute to the understanding of the challenges of vaccination roll-out by examining the motivations, hesitancies, and their associated factors, in obtaining the COVID-19 vaccines two years into the pandemic. Methods: Cross-sectional online surveys were conducted in Norway, the USA, the UK, and Australia (N = 1649). The participants self-reported whether they had obtained one of the COVID-19 vaccines. Those who had obtained a vaccine reported the reason for their motivation, and those who had not obtained a vaccine reported the reason for their hesitancies. Results: More than 80% of the total sample obtained a COVID-19 vaccine because of public health recommendations and trusted that it was safe. Among those who had not obtained one, the most frequent reason was concerns about side effects. Most who obtained the vaccine reported that they believed in science, but many of those who had not obtained one reported distrust. Among those who had not obtained a vaccine, reports of distrust in policies and science were frequent. Concerns about side effects were more common in males and those with lower education, and those living in rural or remote areas. Conclusion: People who endorsed the vaccine believed that the vaccine reduces the risk of illness, protects the health of others, and had trust in scientific vaccination research. Conversely, the most frequent reason for vaccine hesitancy was concerns about side effects, followed by distrust in healthcare and science. These findings could inform public health strategies that aim to increase vaccination rates.

## 1. Introduction

The COVID-19 pandemic has had major public health impacts at all levels, including the individual, local community, national, and international levels. Government departments in public health have provided recommendations and guidelines to follow in order to prevent the spread of coronavirus among their communities. One of the universal measures was the introduction of COVID-19 vaccines; however, there has been a history of vaccine hesitancy observed globally, with it reported in over 90% of countries in the world [1].

Vaccines are important to lower the spread of the virus, although some individuals are hesitant in taking the recommended vaccine advice [2]. Vaccine hesitancy is not only apparent in specific extreme sub-populations, but it has also been shown in healthcare professionals, including nurses and medical workers—particularly those not working with COVID-19 patients [3]. In many countries, public health and medical research has been working to identify the attitudes and perceptions in certain demographics that are preventing them from obtaining the COVID-19 vaccine.

An earlier study by Price and colleagues found that living in a city, having had a college education, being concerned about your health and your next of kin, and trusting information provided by the authorities increased the likelihood of reporting willingness to obtain the COVID-19 vaccine [4]. This study was conducted nine months after the outbreak was declared a pandemic. Since then, the vaccination rates have improved in some nations around the world, although they may not have been across all sub-populations. Higher levels of vaccine hesitancy have been associated with lower vaccination rates and have been linked to distrust in the healthcare policies set by governments, concerns about the side effects, time taken to develop the vaccine and the belief that there was limited inclusion of diverse populations in the initial studies [5,6,7,8,9]. It was recommended that further research is needed to better understand the socio-demographic profiles of people who are not taking up the vaccines and the reasons behind hesitancies in obtaining the COVID-19 vaccine.

Since the beginning of the pandemic, studies have been conducted in different parts of the world to examine the reasons for not obtaining the COVID-19 vaccine. A cross-sectional study from Pakistan found that the major reasons for vaccine refusal were the belief that COVID-19 was not a real problem, that it was a conspiracy, and simply that they do not need the vaccine [10]. In addition, a US study conducted in 2021 reported a fear of side effects, not trusting the vaccine, and not trusting the government as common reasons for vaccine hesitancies [11]. Overall, the common reasons for vaccine refusal reported by studies included being against vaccines in general, concerns in safety, and distrust in the vaccine’s effectiveness [9].

Vaccine hesitancies could differ between individuals. For example, an Italian national survey found that vaccination refusal was more common in females and in individuals living in rural areas [12]. Vaccine hesitancies associated with living outside of city areas and lower levels of education have previously been reported in the Polish population [13]. A UK household population study also reported that vaccine hesitancy levels were higher in people with lower education levels [14]. A Japanese study reported higher vaccine acceptance rates among younger people, and that hesitancy rates were higher among females and individuals with lower education [15]. Vaccine acceptance may also change depending on whether someone has contracted the infection. An Israeli cross-sectional study showed that a previous history of COVID-19 was not associated with increased vaccine hesitancy [16]. However, the association between these individual factors and their association with vaccine hesitancies need to be replicated or studied in different populations and settings.

Previous studies have shown that vaccine hesitancy can be due to many factors, and these factors may vary across different periods and situations [17]. Therefore, we need research from updated resources and across different settings and cultures to increase the knowledge in order to help identify differences and consistencies among those who are motivated or hesitant in obtaining one of the COVID-19 vaccines. Although several studies have examined the overall levels of vaccine hesitancy and its associations, there has been less research focusing on the specific reasons for vaccine hesitancy among individuals who express hesitation. Some previous studies have examined vaccine hesitation in whole populations, including those who have already obtained the vaccine. However, hesitancies may not directly translate to behavior, for example, a population-based cross-sectional study in Turkey found a very high vaccination rate (93.4%), despite the majority of the population expressing vaccine hesitancy (58.4%) [18]. Therefore, it is more informative to focus on the reasons behind hesitancy among those who have not obtained a vaccine. Greater knowledge regarding their reasons may have implications for public health policies and practices targeting hard-to-reach population segments.

This study aimed to examine the factors behind motivations and hesitancies in obtaining the COVID-19 vaccines two years into the pandemic in Norway, the USA, the UK, and Australia. While previous studies have focused on similar topics since the start of the pandemic to the present, our study is novel due to its multi-country approach. In addition, we aimed to identify the socio-demographic factors that are associated with the reasons for hesitancies in obtaining the COVID-19 vaccines. By taking this multi-country approach and examining the individual risk factors, we aimed to contribute to the literature by providing global scientific knowledge to inform public health policies on motivation and hesitancy in obtaining COVID-19 vaccines.

## 2. Method

### 2.1. Study Design and Setting

We conducted a cross-sectional survey two years after the start of the COVID-19 pandemic in four countries (Australia, Norway, the United Kingdom, the United States of America) [19,20]. The cross-sectional survey was disseminated to the general public through social media (e.g., Facebook and Twitter) between November 2021 and January 2022. There was no minimum sample size and we aimed to recruit as many participants as possible during the data collection period. The social media posts were distributed through the researchers’ professional social media accounts and paid advertisements. Therefore, it is not a representative sample.

### 2.2. Participants

Participants met the inclusion criteria if they were 18 years of age or older and understood the language in which the survey was presented (Norwegian or English). There was a total of 1649 (Norway n = 242, UK n = 255, USA n = 915, Australia n = 237) participants who completed the survey.

### 2.3. Measures

#### 2.3.1. Vaccine Motivation and Hesitancies

Participants were asked if they had obtained one of the available COVID-19 vaccines. Those who reported ‘yes’ were asked to report the reason for their motivation, and those who reported ‘no’ were asked to report the reason for their hesitancies. The specific motivation and hesitancy response options are described below.

Among the participants who reported receiving the COVID-19 vaccine, they were subsequently asked about their motivations for obtaining the vaccine. They were presented with a list of motivators and asked to select all that applied. The list of motivators included: (1) Vaccines reduce risks of illnesses; (2) The vaccine is safe, with no concerns about side effects; (3) Employers required it; (4) Health of others; (5) Believe in the science behind the vaccine development; (6) Public health recommendations; (7) Other (If other, please add your reasons).

Participants who reported not obtaining a COVID-19 vaccine were asked if their hesitancy to obtain the COVID-19 vaccine was related to: (1) Distrust in healthcare policies; (2) Distrust in science; (3) Concerns about side effects; (4) Religious beliefs related to vaccines; (5) Other (If other, please add your reasons).

#### 2.3.2. Socio-Demographic Variables

The socio-demographic variables measured included age (18–29, 30–39, 40–49, 50–59, 60+), gender (female, male, other/prefer not to say), the highest education level (vocational or lower, bachelor’s degree or higher), employment status (employed, not employed, other), marital status (have a spouse or partner, or not), area of residence (rural/remote, town/suburb, city/metropolitan areas), and previous COVID-19 infection (yes/no).

### 2.4. Statistical Analysis

We compared the proportion of participants who had and who had not obtained a COVID-19 vaccine in the overall sample and compared the differences by country. We plotted the percentages of participants endorsing each of the motivational and hesitancy reasons.

Among those who had not obtained a COVID-19 vaccine, the reported reasons were cross-tabulated by socio-demographic factors to examine their bivariate relationship using Chi-Square tests. Then, multiple logistic regression models were used to examine the sociodemographic associates for each of the vaccine hesitancy reasons. All of the independent variables were entered in one step, including country, age, gender, education level, employment status, and marital status. The outcome variables were reasons for not obtaining the vaccine. Adjusted odds ratios (OR) were reported as the effect size, and the 95% confidence interval of the OR was reported. Statistical significance was set at *p* < 0.05.

### 2.5. Ethics

This study received ethical approval from the review boards at the universities where the study was conducted: OsloMet (20/03676) and the regional committees for medical and health research ethics (REK; ref. 132066) in Norway; the University of Michigan Institutional Review Board for Health Sciences and Behavioral Sciences (IRB HSBS), which designated the study as exempt (HUM00180296) in the USA; the University of Central Lancashire (Health Ethics Review Panel) (HEALTH 0246) in the UK; and the University of Queensland Human Research Ethics Committees in Australia (HSR1920-080 2020000956).

## 3. Results

### 3.1. Descriptive Statistics

At the time of the survey, overall, 84.7% (n = 1396) of the participants had obtained one of the available COVID-19 vaccines, and 15.3% (n = 253) had not. Among the participants, the highest proportion of individuals who had not obtained a vaccine was observed among the USA participants (18.7% had not), while the UK participants had the lowest proportion (5.5% had not). Norway (14.0% had not) and Australia (14.3% had not) fell in between (see Table 1).

There was a spread of participants across the age groups, with lower proportions of USA participants aged over 50, but a higher proportion of older participants in the Australian sample. Over 70% of the sample were female; in Australia, it was 81%. The participants predominantly included people with higher education, who were employed, and the majority had a spouse or partner. Most participants lived in either a town, suburban area, or in a city, and had not been infected with COVID-19 before.

### 3.2. Frequency of Reasons for Having and Not Having Obtained the COVID-19 Vaccines

Among those who had obtained the vaccine (n = 1396), the most common reasons that motivated them to do so were that vaccines reduce the risk of illnesses (86%), the health of others (74%), and belief in science (71%; see Figure 1). Over half of the participants also endorsed that they were motivated by public health recommendations (66%) and that it was safe, with no concerns about side effects (56%). A low, but substantial, proportion of the participants reported that they received the vaccine because their employer required it (15%).

Among those who had not obtained the COVID-19 vaccines (n = 253), the top reasons for their hesitancy were concerns about the side effects (79%) and distrust in healthcare policies (67%). Distrust in science was reported by 45% and religious beliefs related to vaccines were reported by 27% of the participants. Other low-frequency reasons included: believing the vaccine was not necessary (6%) and wanting to resist control or coercion of the authorities (9%).

### 3.3. Socio-Demographic Associates of Vaccine Hesitancy

A crosstabulation of the bivariate associations between the vaccine hesitancy reasons by socio-demographic factors is presented in Table 2. Between the participants of the four countries, there were no significant differences in the endorsements of the reasons reported, except that the USA participants were the most likely (36.3%), and the Norwegian participants were the least likely (2.9%), to report that they had not obtained a COVID-19 vaccine due to religious beliefs. In addition, younger participants, aged 18–29, were less likely to report that their hesitation was due to religious beliefs. Reports of distrust in healthcare policies increased with age, with 82.5% of those aged 60 and older endorsing this reason. Concerns about side effects were most commonly reported by females (84.1%) and those who were employed (82.0%). In addition, the proportion of participants endorsing the reasons for vaccine hesitancy were higher in the rural population and among those with a previous COVID-19 infection.

The multiple logistic regression results of each of the vaccine hesitancy reasons by the socio-demographic factors are presented in Table 3. Compared to the USA participants, those from Norway and the UK had significantly lower odds of reporting religious beliefs as their reason for not having obtained one of the COVID-19 vaccines. The UK participants had lower odds of reporting distrust in healthcare policies or science and concerns about the side effects compared to the USA participants. The Norwegian participants who had not obtained a COVID-19 vaccine had significantly higher odds of reporting distrust in science as their reason than the USA participants. The participants in the 18–29 age group and those with lower education had significantly lower odds of reporting all of the vaccine hesitancy reasons. Males had higher odds of reporting distrust and concern reasons, and participants of other—or preferred not to say—genders had higher odds of reporting that their vaccine hesitancy was related to distrust in healthcare policies. Those living in a rural or remote area had higher odds on almost all of the reasons for vaccine hesitancy—apart from religious beliefs—than any other area of residence. The participants who were previously infected also had higher odds of reporting all of the reasons—apart from distrust in science—for vaccine hesitancy compared to those who had not been infected.

## 4. Discussion

Among those who had obtained the vaccine, the most frequently endorsed reasons were that vaccines reduce the risk of illness and that they protect the health of others. Over half obtained a COVID-19 vaccine because of public health recommendations and trusted that it was safe, while among those who had not obtained one, the most frequent reason was concerns about side effects. Vaccine safety concerns had been a main reason for vaccine hesitancy even before the COVID-19 pandemic [1].

Vaccine uptake has a large impact on public health because many had become seriously ill, and many died, due to COVID-19, particularly during the early stage of the pandemic, before vaccines were available. However, there were in fact cases of serious side effects related to receiving the developed COVID-19 vaccine [21], which have been widely reported. It was hoped that the transparency about these adverse events would inspire trust in the subsequent process of developing and approving new vaccines for public distribution and use. It is possible that the massive media attention concerned with the early cases of severe illness and death after receiving the AstraZeneca vaccine resulted in an increased fear of side effects among specific populations. Indeed, this resulted in the vaccine not being approved for use in the USA, as public awareness and concerns were high, which may have impacted the overall hesitancy of obtaining any vaccines.

The current study found that most of those who obtained the vaccine reported that they believed in science, and on the contrary, many of those who had not obtained the vaccine reported that it was because they distrusted healthcare policies and science. There was some overlap between the various listed reasons for hesitating to obtain the vaccine. While some distrust in science may have been accentuated due to the media attention concerned with the early AstraZeneca vaccine, a proportion of those not vaccinated indicated several reasons for their hesitancy. Thus, among those indicating several reasons, the hesitancy to obtain the COVID-19 vaccine may be considered one aspect of a more general tendency to distrust authorities [9]. The previous results of Price and colleagues also support the link between a general trust in public authorities and willingness to obtain the COVID-19 vaccine [4].

We found that a small but substantial proportion (15%) reported to have obtained the vaccine because their employer required it. Employer requirement is an external motivation, meaning people may obtain the vaccine even if they are hesitant to do so to avoid facing job loss. Health authorities collaborating with businesses and employers may help to increase the vaccination rates in the population in a time of crisis. It is crucial to consider the role of country policies in shaping vaccination practices. In some countries, such as the United States, vaccination mandates from universities and employers have been implemented to ensure the safety and well-being of students and staff. These policies reflect a broader public health approach that aims to achieve high vaccination coverage and mitigate the spread of infectious diseases. However, it is important to note that vaccination policies can vary significantly between countries due to differences in the healthcare systems, legal frameworks, and cultural contexts. While our study primarily focuses on individual perspectives, we acknowledge the influence of country policies on vaccination behaviors and outcomes across different settings. Future research could explore the interplay between individual attitudes, country policies, and vaccination uptake to provide a comprehensive understanding of the factors that shape vaccination practices in different contexts. Such investigations can contribute to the development of targeted interventions and policy recommendations to enhance vaccination coverage and promote public health.

Within our study, the vaccination rate was highest among the UK participants and lowest among the US participants, and among those who had not obtained a vaccine, the US sample was more likely to endorse all of the vaccine hesitancy reasons in general. A study using a US sample exploring vaccine hesitancy saw similar results across income/employment levels and age [22]. However, they did not investigate the impact of an individual’s religious beliefs on the choice to vaccinate, which is an important determinant, particularly in the US setting. Our study found that, overall, among people who had not obtained a COVID-19 vaccine, approximately one-third reported that it was due to religious beliefs related to vaccines. Indeed, the comparisons by country showed that the US participants were much more likely to endorse that they had not obtained a COVID-19 vaccine due to religious beliefs than, for example, the Norwegian participants. This result may be associated with misleading information being spread to religious communities on social media [23].

We identified some socio-demographic associations with the reasons for not having obtained a COVID-19 vaccine. Among those who had not obtained a COVID-19 vaccine, reports of distrust in policies and science and concerns about the side effects were more common in males, those with lower education, and people living in rural or remote areas. Public health strategies that aim to increase vaccine uptake rates may consider socio-demographic differences in the design of interventions. A previous study that was conducted between October and November 2020 found that, across all countries, trust in the information provided by public authorities was associated with a willingness to obtain the COVID-19 vaccine [4]. It had been identified that those who had lower levels of education were less likely to be willing to obtain the COVID-19 vaccine at that time, and this was consistent in our current study. This implies that public health interventions to promote the uptake of COVID-19 vaccines may not have been successful in reducing vaccine hesitancy among those with lower levels of education.

In regard to gender differences, our results were consistent with the earlier studies in which males were more likely to report that they were unwilling to obtain the vaccine [4]. However, there have been mixed findings on gender differences in the willingness to obtain vaccines. For example, a UK study found that women were more likely to be vaccine-hesitant [24]. Higher rates of vaccination refusal among females have also been reported in other countries, including Italy and Japan [12,15]. A consistent finding across different settings and times was that distrust is still a leading contributor to vaccine hesitancy and that this issue needs to be addressed to increase vaccine uptake. Gender differences in vaccine acceptance are important to consider because different gender roles in different cultures and settings may be the impacting factor for the vaccination status within families. Although our study was conducted in four Western countries with non-representative samples, some reasons for vaccine hesitancies may apply universally. For example, in line with our findings, an Ethiopian study also reported a lack of trust in the vaccine, doubts regarding the side effects, and religious reasons as common refusal reasons [25]. A polish population study reported that individuals living in cities and those with higher education had higher levels of COVID-19 vaccine acceptance, which were also in line with our findings [13].

Potential negative reactions to the vaccine is considered as a reason for concern by many as we found a large proportion of our participants indicated that they were concerned about side effects. Concerns about the health effects of the vaccines have been an ongoing issue that is associated with willingness to obtain the COVID-19 vaccine [4]. Concerns about the health effects can both increase and decrease the vaccine uptake rates. The vaccine-hesitant group are concerned that the vaccine would impact their health, while the vaccine-motivated group believe that the vaccine will protect them from the harms of COVID-19. This shows that a pathway to increasing vaccine uptake is to increase the trust in policy and health authorities on the benefits of vaccines that would outweigh the potential negative side effects. To address vaccine hesitancy and to protect public health, collaboration is needed between governments, private companies, religious groups, and the community to promote public trust of vaccines [26].

### 4.1. Limitations

The following are the key limitations of this study. The survey had not specified hesitancy in obtaining a certain dose of the COVID-19 vaccine. The hesitancies and reasons for obtaining the first dose may be different to the successive booster doses. For example, a repeated cross-sectional survey in Hong Kong found that vaccine hesitancies were higher in the third COVID-19 wave than the first wave [27]. Secondly, our list of reasons presented in the survey was not exhaustive as to why an individual would choose to not obtain the vaccine. To explore additional potential reasons, we provided participants with an option to provide open-ended responses if they had any other reasons. We did not identify any other major high-frequency reasons, but the main other low-frequency reasons were believing the vaccine was not necessary and wanting to resist control or coercion of the authorities, which may be considered by public health campaigns aiming to promote vaccine uptake.

Another limitation is that our surveys were only undertaken in four selected developed countries. We have not conducted any surveys in less developed countries. Given the significant differences in cultural ways of living, our findings should not be generalized to other countries with large cultural differences. COVID-19 has also had differential impacts on mental health [28], but this study has not provided any investigations into potential mental health impacts on the motivations and hesitancies in obtaining COVID-19 vaccines. Similarly, social-economic status was not examined. To address these limitations, future cross-national surveys on the same topic should simultaneously be undertaken in other settings and with other variables of interest.

There are limitations related to our data collection method. As we collected the data through an online sample through social media distribution, it is unlikely to be representative of the general population. Further, we had larger sample sizes from the USA recruitment, and smaller samples from the other three countries. A contributing factor to the smaller sample sizes from Australia, the UK, and Norway was the availability of participants within the targeted population. As our recruitment strategy primarily relied on social media platforms, it can introduce certain biases in participant selection. Despite our efforts to target diverse populations, it is possible that the reach and effectiveness of our recruitment methods varied across countries. Local factors, such as the popularity of specific social media platforms, have probably influenced the recruitment outcomes.

While we acknowledge the concern about data credibility in surveys conducted through social media platforms, we believe that the potential bias is likely to be comparable to other self-report surveys not conducted using social media. Participation in our study was voluntary and without any incentives, minimizing the likelihood of intentional misrepresentation. Previous studies that have recruited participants using Facebook have demonstrated that it could be an effective method to gather survey data and could inform policy options for targeting high-hesitancy groups [11].

The higher proportion of female respondents in our survey reflected the demographic distribution on the social media platforms used for recruitment and self-selection for participation bias. While we acknowledge the gender imbalance in our sample, our logistic regression results were adjusted for gender, ensuring that the analysis accounts for potential gender-related differences. We acknowledge the need for further efforts to increase male representation in future studies to capture a more balanced perspective.

Despite the fact that our sample is unlikely to be representative of the general population, our findings on the sociodemographic factors associated with vaccine hesitancies are likely to apply to the general population. However, our results on the proportion of people endorsing each of the vaccine hesitancy reasons may not. Indeed, the vaccination rates in our sample were different from those reported in the population of each country. Our samples in the UK (sample: 94.5%, population level: 88.9%) [29], Norway (sample: 86%, population level: 79.5%) [30] and the USA (sample: 81.3%, population level: 75.1%) [31] had a higher first dose vaccine rate compared to the general population levels. The implications of this could be that our sample included more of those in the population who have more positive views towards vaccination than the general population. Another reason that our sample recorded a higher vaccine rate could be due to the social desirability bias that is present when covering COVID-19 vaccination status [32,33]. However, our top reasons for vaccine uptake and hesitancy were consistent with previous single country studies, e.g., the UK population ‘Understanding Society’ COVID-19 survey of over ten thousand participants [14]. Our study contributes to the existing literature by providing findings across four countries using standardized methodologies and an investigation of additional socio-demographic variables associated with vaccine hesitancies. At the time of the survey, the type of vaccines available were Pfizer, Moderna, AstraZeneca, and Novavax. In our study, we do not know if the hesitancies in obtaining the vaccines depended on the brand of vaccine, which warrants future research.

Despite having a large sample of those who had not obtained a vaccine in our USA sample (n = 171, 18%), the sample size of participants who had not obtained the vaccine was small in Norway (n = 34, 14%), the UK (n = 14, 5%), and Australia (n = 34, 14%). However, taking the total participants from each country into account, this skewness is explainable, and apart from the low rate in Australia, the differences were not of significance. However, future research could actively recruit participants who choose to not obtain the vaccine and assess their reasons for not opting to get vaccinated. This could lead to potential breakthroughs in strategies that could help encourage those people to get vaccinated. Another limitation linked to our data collection method was that our motivational factors may be prone to post-hoc rationalization, although our hesitancy factors were still important for understanding which factors were key for those who have chosen not to obtain the vaccine.

### 4.2. Conclusions

The study assessed the socio-demographic profiles, reasons for obtaining the COVID-19 vaccine, and reasons why individuals may be hesitant across the UK, the USA, Norway, and Australia, two years after the start of the COVID-19 pandemic. The study broadened the ongoing global discourses on the motivations and hesitancies in obtaining COVID-19 vaccines. The participants who endorsed the vaccine believed that the vaccine reduces the risk of illness, protects the health of others, and had trust in scientific vaccination research. Conversely, the most frequent reason for vaccine hesitancy was concerns about side effects, followed by distrust in healthcare and science. Demographically, hesitancy was more prevalent among males, those with lower education, and people living in rural or remote areas. Understanding the demographics and variables linked to vaccine hesitancy will help enrich future healthcare practice, policy development, and program implementation. Therefore, appropriate methods will be used to increase vaccination uptake. These findings contribute to the existing scientific knowledge by broadening the ongoing global discourses on the motivations for and hesitancies of obtaining COVID-19 vaccines, which could be used to inform global strategies for promoting vaccination uptake.

## Figures and Tables

**Figure 1 vaccines-11-01086-f001:**
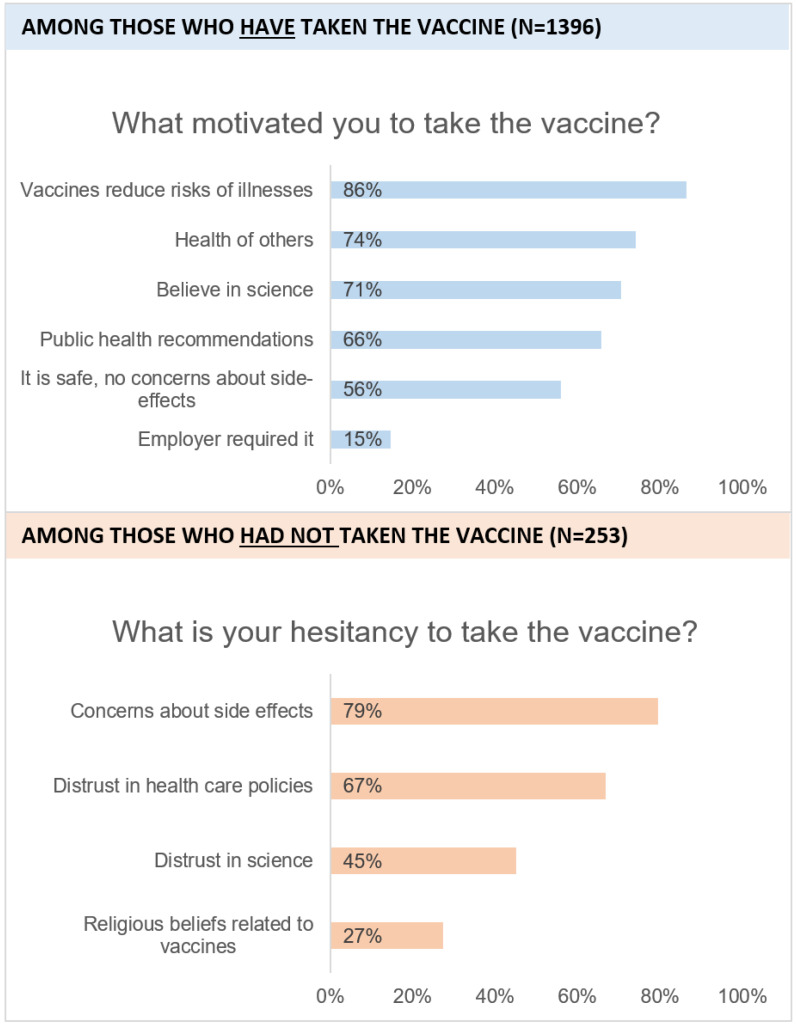
Frequency of motivation and hesitancy reasons for having and not having obtained the COVID-19 vaccines.

**Table 1 vaccines-11-01086-t001:** Descriptive statistics of participants from Norway, the UK, the USA, and Australia (N = 1649).

		Column Percentages (%)				
		Norway	UK	USA	Australia	*p*
		(n = 242)	(n = 255)	(n = 915)	(n = 237)	
Have obtained a COVID-19 vaccine						
	No	14.0	5.5	18.7	14.3	<0.001
	Yes	86.0	94.5	81.3	85.7	
Age groups						
	18–29	13.6	13.3	19.8	8.9	<0.001
	30–39	24	22	32.3	6.3	
	40–49	21.1	29.8	33.2	11.4	
	50–59	24.4	26.3	8.3	22.8	
	60+	16.9	8.6	6.3	50.6	
Gender						
	Female	77.7	77.6	72.6	81.0	0.001
	Male	20.7	20.8	21.2	16.5	
	Other/prefer not to say	1.7	1.6	6.2	2.5	
Education level						
	Vocational or lower	21.1	25.1	21.3	34.2	<0.001
	Bachelor’s or higher	78.9	74.9	78.7	65.8	
Employment						
	Employed	79.8	83.5	75.5	37.6	<0.001
	Not employed	14	8.2	13.2	27.8	
	Other	6.2	8.2	11.3	34.6	
Spouse or partner						
	Yes	62.8	68.6	67.0	51.5	<0.001
	No	37.2	31.4	33.0	48.5	
Area of residence						
	Rural or remote	8.7	17.6	19.6	1.7	<0.001
	Town or suburban	35.5	62.4	55.7	25.3	
	City or metropolitan	55.8	20	24.7	73.0	
Previous infection						
	Yes	5.4	29	24	1.3	<0.001
	No	94.6	71	76	98.7	

**Table 2 vaccines-11-01086-t002:** Crosstabulation of vaccine hesitancy reasons by socio-demographic factors among those who had not obtained a COVID-19 vaccine.

		n	Reason for Not Having Obtained One of the COVID-19 Vaccines, among Those Who Had Not (n = 253)							
			Distrust in Healthcare Policies		Distrust in Science		Concerns about Side Effects		Religious Beliefs Related to Vaccines	
			%	*p*	%	*p*	%	*p*	%	*p*
Country										
	Norway	34	61.8	0.523	61.8	0.078	85.3	0.185	2.9	<0.001 **
	UK	14	64.3		35.7		92.9		14.3	
	USA	171	63.7		38.0		73.1		36.3	
	Australia	34	76.5		41.2		79.4		11.8	
Age groups										
	18–29	26	42.3	0.022 *	30.8	0.742	65.4	0.257	7.7	0.010 *
	30–39	65	66.2		43.1		83.1		32.3	
	40–49	90	64.4		40.0		80.0		35.6	
	50–59	32	62.5		46.9		71.9		28.1	
	60+	40	82.5		45.0		70.0		12.5	
Gender										
	Female	138	63.8	0.243	45.7	0.332	84.1	<0.001 **	31.9	0.193
	Male	89	70.8		37.1		74.2		21.3	
	Other	26	53.8		34.6		46.2		23.1	
Education level										
	Lower	109	69.7	0.190	45.0	0.332	75.2	0.635	29.4	0.517
	Bachelor’s or higher	144	61.8		38.9		77.8		25.7	
Employment										
	Employed	161	66.5	0.786	41.6	0.766	82.0	0.029 *	29.2	0.659
	Not employed	38	60.5		36.8		65.8		23.7	
	Other	54	64.8		44.4		68.5		24.1	
Spouse or partner										
	Yes	182	64.8	0.838	43.4	0.325	78.0	0.419	29.7	0.170
	No	71	66.2		36.6		73.2		21.1	
Area of residence										
	Rural or remote	68	17.7	<0.001 **	11.6	0.003 *	18.5	<0.001 **	8.4	<0.001 **
	Town or suburban	127	9.4		6.7		12.5		4.3	
City		58	8.2		5.1		9.1		2.2	
Previous COVID-19 infection										
	Yes	94	18.1	<0.001 **	9.7	0.033 *	21.9	<0.001 **	9.4	<0.001 **
	No	159	8.4		6.7		9.9		3.0	

Note: * *p* < 0.05, ** *p* < 0.01.

**Table 3 vaccines-11-01086-t003:** Logistic regression on specific vaccine hesitancy reasons by socio-demographic factors.

		Reason for Not Having Obtained One of the COVID-19 Vaccines, among Those Who Had Not Obtained One							
		Distrust in Healthcare Policies		Distrust in Science		Concerns about Side Effects		Religious Beliefs	
Sociodemographic Variables		OR [95%CI]	*p*	OR [95%CI]	*p*	OR [95%CI]	*p*	OR [95%CI]	*p*
Country (ref: USA)									
	Norway	1.10 [0.63–1.92]	0.745	1.94 [1.07–3.48]	0.028 *	1.47 [0.90–2.40]	0.128	0.08 [0.01–0.58]	0.013 *
	UK	0.24 [0.11–0.49]	<0.001 **	0.35 [0.15–0.80]	0.012 *	0.32 [0.17–0.58]	<0.001 **	0.08 [0.02–0.35]	<0.001 **
	AUS	1.48 [0.79–2.76]	0.217	1.64 [0.79–3.40]	0.184	1.91 [1.06–3.43]	0.03 *	0.35 [0.10–1.19]	0.093
Age groups (ref: 40–49)									
	18–29	0.24 [0.12–0.49]	<0.001 **	0.37 [0.17–0.81]	0.013 *	0.32 [0.18–0.58]	<0.001 **	0.09 [0.02–0.40]	0.001 **
	30–39	0.79 [0.50–1.24]	0.299	0.93 [0.55–1.58]	0.786	0.85 [0.57–1.28]	0.441	0.75 [0.41–1.36]	0.338
	50–59	0.62 [0.34–1.12]	0.114	0.81 [0.42–1.59]	0.545	0.60 [0.35–1.03]	0.062	1.07 [0.46–2.45]	0.881
	60+	0.82 [0.46–1.47]	0.504	0.85 [0.43–1.67]	0.629	0.60 [0.34–1.05]	0.073	0.38 [0.13–1.15]	0.088
Gender (ref: female)									
	Male	2.87 [1.97–4.19]	<0.001 **	1.82 [1.15–2.87]	0.01 *	2.16 [1.52–3.08]	<0.001 **	1.40 [0.77–2.56]	0.274
	Other	2.59 [1.30–5.16]	0.007 *	2.10 [0.94–4.69]	0.07 *	1.52 [0.75–3.07]	0.248	1.51 [0.57–4.02]	0.413
Education (ref: higher)									
	Lower	3.41 [2.39–4.85]	<0.001 **	3.15 [2.08–4.76]	<0.001 **	2.94 [2.11–4.12]	<0.001 **	3.55 [2.07–6.07]	<0.001 **
Employment (ref: employed)									
	Not employed	1.01 [0.60–1.68]	0.978	1.13 [0.62–2.03]	0.697	0.98 [0.61–1.56]	0.918	1.11 [0.51–2.42]	0.785
	Other	1.79 [1.10–2.92]	0.019 *	2.13 [1.24–3.66]	0.006 *	1.52 [0.96–2.41]	0.075	1.78 [0.87–3.66]	0.115
Partnered (ref: yes)									
	No	0.80 [0.54–1.17]	0.253	0.63 [0.40–1.00]	0.050	0.71 [0.50–1.02]	0.066	0.68 [0.36–1.27]	0.221
Area of residence (ref: city)									
	Rural/remote	2.20 [1.30–3.72]	0.003 *	2.60 [1.40–4.84]	0.003 *	2.08 [1.26–3.43]	0.004 *	1.81 [0.82–3.98]	0.141
	Town/suburb	1.28 [0.83–1.96]	0.267	1.59 [0.96–2.64]	0.073	1.58 [1.06–2.35]	0.024 *	1.301 [0.65–2.62]	0.461
Previous COVID-19 infection (ref: no)									
	Yes	2.26 [1.51–3.40]	<0.001 **	1.57 [0.95–2.58]	0.078	2.58 [1.78–3.74]	<0.001 **	2.23 [1.29–3.87]	0.004

Note. OR: odds ratio; 95%CI: 95% confidence intervals. * *p* < 0.05, ** *p* < 0.01.

## Data Availability

Data available on request due to ethical restrictions.

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
