# Peer review of "Motivation and Hesitancies in Obtaining the COVID-19 Vaccine—A Cross-Sectional Study in Norway, USA, UK, and Australia"

_vaccines, 2023, doi:10.3390/vaccines11061086_

Round 1
Reviewer 1 Report
I have reviewed the manuscript entitled “Motivation and hesitancies in taking the COVID-19 vaccine – a cross-sectional study in Norway, USA, UK, and Australia” and provide my comments below. In general, the study’s background is poorly presented, and the study aims are not rationalized well. The Authors support their Introduction and Discussion with a very low number of references. Overall, there are only 18 references cited, while the volume of literature on COVID-19 vaccine hesitancy is huge. Some methodological issues are not well explained, and there are also some significant methodological deficiencies. In conclusion, there are numerous concerns that prevent me from recommending this manuscript for publication.
Language:
To my surprise, since the study was also conducted in English-native-speaking countries, the manuscript's language is far from optimal. There are parts with unfinished sentences, parts that are technically not a sentence at all, or which are hard to follow. Some examples:
1. Line 9-10: Those who had reported the reason for their motivation, and those who had not reported their reason for hesitancies. – this is not properly phrased.
2. Line 19-20: Public health strategies that aim to increase vaccination. – this seems incomplete.
3. Line 22-23: The COVID-19 pandemic has had major public health impacts on all levels, from an individual level, local community level, national level, and international level. – very repetitive language.
4. Line 51-54: Although there have been several studies that examined overall levels and associates of vaccine hesitancy, there has been less research focusing on the specific reasons for vaccine hesitancy among those who hesitate. – hard to follow and repetitive
5. Line 253-254: To understand potential additional reasons, we included an additional option for participants to write their open-ended responses, if they had any other reasons. – again, repetitive.
6. Line 79-81: Among participants who had recorded taking the COVID-19 vaccine, they were subsequently asked what motivated them to take the vaccine. They were then presented with a list of motivators and were asked to select all that apply. – this is hard to follow.
7. Line 120-123: The USA participants (18.7% had not) had the highest proportion who had not taken a vaccine, while the UK participants (5.5% had not) had the lowest proportion, and Norway (14.0% had not) and Australia (14.3% had not) were in between (see Table 1). – this is technically very poor language.
Intoduction:
Line 43-46: In this part: „Due to these deficits in knowledge previous studies recommended that further research is needed to better understand the socio-demographic profiles of people who are not taking up the vaccines and the reasons behind hesitancies in taking the COVID-19 vaccine”, the authors refer to selected papers. Firstly, there are numerous studies omitted by the authors, if they would be acknowledged, maybe the authors would not state that there are “deficits in knowledge”. Secondly, even using the referenced work, it would be unsupported to state that there are knowledge gaps.
In general, the rationale for this study and its aims is poorly outlined. Much more work would be necessary to prepare the background, use the existing body of knowledge, and identify the potential gaps. In other words, the Introduction is far from satisfactory.
Methods:
2.1. Study design and setting:
Line 65-67: The explanation of how the survey was disseminated is poor. The authors state: “The cross-sectional survey was disseminated to the general public through social media (e.g., Facebook, and Twitter) between November 2021 and January 2022.” This is an imprecise description. Where exactly was the invitation posted? Through your personal social media accounts? Using social media to survey is far from satisfactory and is prone to a number of biases.
2.2. Participants:
Was your sample representative? Did you assess it? The methodology provides no explanation on this matter, while it is a pivotal parameter for surveys. Was there any minimum sample size assessed prior to the study?
The sample size is relatively small; apart from the USA, there is approx. 250 individuals were surveyed in Norway, UK, and Australia. This cannot be representative, particularly given the methodology to disseminate the survey through social media. In general, the sample size in the US is equal to 0.0003% of the population, in the UK – 0.0004%, in Australia – 0.0009%, and in Norway – 0.005%. Given all this, it is hard to understand how this study could fill any “knowledge deficits”.
2.3 Measures
2.3.2 Socio-demographic variables
There are some important factors that could influence the results but were not included in the study, such as place of living (urban vs. rural areas) and economic status. According to other studies, they may affect vaccine hesitancy.
Line 97-98: Percentages of participants endorsing motivational and hesitancy reasons were plotted to identify the most common reasons. – this should be rephrased.
3. Results
3.2. Frequency of reasons for having and not having taken the COVID-19 vaccines
Line 133-143: You just repeat all the data given in Figure 1. This is inappropriate and boring. Please use Figure to provide details and text to provide some generalized observations.
You have used the option “others” for motivations and hesitancy but did not specify what other reasons were given.
4. Discussion
The Discussion repeats too much of the reported results and fails to fully discuss them while referring to other studies. The authors refer only to 4 other studies, while there is a volume of literature to use.
There is a long subsection on Limitations, which clearly shows that the Authors understand the significant deficiencies of their study.
Line 254-258: This belongs to the Results: “We did not identify any other major high-frequency reasons, but we did find other low-frequency reasons including believing the vaccine is not necessary (6%) and wanting to resist control or coercion of the authorities (9%). These factors may be considered by public health campaigns aiming to promote vaccine uptake.”
4.2 . Conclusion
Line 305-307: This is a redundant part: “To better understand the challenges of the COVID-19 uptake, we conducted a cross-sectional online survey to examine the reasons and socio-demographic associates of hesitancies in taking the COVID-19 vaccines two years into the pandemic across four countries.”
The quality of the language is poor. See my other comments.
Reviewer 2 Report
Dear Authors
I find your work here very interesting, to collect responses across four countries in three continents. In the methodology you used a small sample size to all countries other than the USA. I think for a sample to be representative it should be 385 at least. You could explain the limitations that prevented you from getting a representative sample from the 3 countries. Otherwise I think your study was scientifically sound and very informative.
Reviewer 3 Report
My comments are uploaded for the responses from the authors.

Minor english grammatical errors and editing is required to fix them.
Reviewer 4 Report
Known in the field based on previous literatures:
- Covid-19 is an infectious disease caused by severe acute respiratory syndrome coronavirus 2 (SARS-CoV-2). The disease has blowout worldwide, and still leading to an ongoing pandemic.
- Symptoms of COVID‑19 are variable, but often include cough, breathing difficulties, headache, fever, fatigue, loss of smell and taste.
- COVID‑19 vaccine is developed and intended to provide acquired immunity against the virus.
In this article authors reported following findings:
I have gone through the article titled "Motivation and hesitancies in taking the COVID-19 vaccine – a cross-sectional study in Norway, USA, UK, and Australia’. Authors investigated the cross-sectional survey after two year of post COVID-19 pandemic in four countries about motivation and hesitancy in taking the COVID-19 Vaccine. Following are the main points-
- Hesitancy to take the COVID-19 vaccine was related to concerns about side effect, religious belief, distrust in science, lack of education and awareness.
- Motivation to take COVID-19 vaccine was the belief in science and awareness about vaccines that their intake reduces risks of illnesses.
Although the many facts and material presented are already available and there is no new things but authors nicely mentioned many facts related to COVID-19 vaccine hesitancy. The data presented are interesting and generally supportive of the conclusions drawn. The following suggestions if incorporated could help in the better understanding of the significance of the work and implications.
Minor Concerns:
1. Authors should clearly mention about how this article different from rest excluding the online survey in the four developed countries. Does it embrace a specific gap in the field?
2. What was the similarity and dissimilarity of motivation and hesitancy in these four countries? Authors can add a table.
3. Method is not sufficiently described. Please describe more in study design and setting. How did you gather all these things from social media?
Round 2
Reviewer 3 Report
The author has answered all my comments. I am satisfied with the response.
Minor to moderate english grammar may be checked.
Author Response
We thank the reviewer for their final comments. We have re-read our manuscript and have made edits to improve our grammar.